# Sparse Convolutions on Lie Groups

**Tycho F.A. van der Ouderaa**                TYCHO.VANDEROUDERAA@IMPERIAL.AC.UK
**Mark van der Wilk**                                      M.VDWILK@IMPERIAL.AC.UK
*Imperial College London*

**Editors:** Sophia Sanborn, Christian Shewmake, Simone Azeglio, Arianna Di Bernardo, Nina Miolane

## Abstract

Convolutional neural networks have proven very successful for a wide range of modelling tasks. Convolutional layers embed equivariance to discrete translations into the architectural structure of neural networks. Extensions have generalised continuous Lie groups beyond translation, such as rotation, scale or more complex symmetries. Other works have allowed for relaxed equivariance constraints to better model data that does not fully respect symmetries while still leveraging on useful inductive biases that equivariances provide. How continuous convolutional filters on Lie groups can best be parameterised remains an open question. To parameterise sufficiently flexible continuous filters, small MLP hypernetworks are often used in practice. Although this works, it typically introduces many additional model parameters. To be more parameter-efficient, we propose an alternative approach and define continuous filters with a small finite set of basis functions through anchor points. Regular convolutional layers appear as a special case, allowing for practical conversion between regular filters and our basis function filter formulation, at equal memory complexity. The basis function filters enable efficient construction of neural network architectures with equivariance or relaxed equivariance, outperforming baselines on vision classification tasks.

## 1. Introduction

Symmetries provide powerful inductive bias in neural networks, often improving data efficiency and generalisation performance. Convolutional filters are an apparent example encoding equivariance to translations through effective weight-sharing. Extensions to other groups exist, but equivariance to Lie groups can be hard to implement efficiently due to their continuous nature. Similar difficulties arise in relaxations of equivariance, which can be useful to model data that only partially respects symmetries. In practice, continuous filters are often parameterised using shallow MLPs, or hypernetworks, which take coordinates (or group elements) as input and output a filter response. Although MLPs can approximate any function in theory, this might require adding a large set of parameters. To prevent having to add many hypernetwork parameters, we propose a basis function approach and parameterise continuous kernels from a small finite set of anchor points. Anchor points are the filter values at specific locations stored in memory and induce values at all other (continuous) locations. In theory this yields an equally flexible model class given enough anchor points, but more crucially, we empirically find that we can represent useful continuous filters with just a few anchor points. We demonstrate that this yields more efficient representations of regular convolutional kernels, group convolutional filters and relaxed convolutional filters. Unlike prior work, this allows us to create group equivariant and relaxed equivariant versions of large commonly used neural network architectures, such as deep ResNets, with equivalent parameter counts. On CIFAR-10 and CIFAR-100 image classification tasks, we show that the approach is much more memory efficient and outperforms our baselines in terms of parameter efficiency and classification performance.

## 2. Related Work

**Continuous convolutional kernels**   Continuous convolutional kernels have been proposed in Schütt et al. (2017) to model quantum interactions, but relied on isotropic kernels limiting flexibility. Wang et al. (2018); Ma et al. (2022) proposed more expressive MLP hypernetworks.

Rahimi et al. (2007) show that the exponential basis function can be approximated (and is exact in the infinite width limit) with a finite number of random Fourier basis functions. Parameterising kernels in such feature spaces using MLPs with sinusoidal activation functions, such as positional encodings (Vaswani et al., 2017) and SIRENs (Sitzmann et al., 2020), have been found effective to fit high-frequency signals, such as images (Tancik et al., 2020) and neural rendings (Mildenhall et al., 2020). In the context of (group-) convolutions such parameterisations are often used to parameterise continuous kernels, and found to be very effective in practice (Romero et al., 2021; Knigge et al., 2022).

Yet, representing filters with MLPs often requires many additional model parameters compared to discrete filter grids used of classical CNNs. We overcome this limitation and effectively parameterise continuous kernels by regressing a small set of anchor points.

**Group equivariance**   Classical convolutional layers embed equivariance to discrete translations. This was generalised to groups in Cohen and Welling (2016) which demonstrated equivariance to discrete 90 degree rotations and flips. Further extensions allow convolutions on other groups, such as continuous roto-translations (Worrall et al., 2017; Weiler et al., 2018; Weiler and Cesa, 2019; Kondor et al., 2018), scale (Worrall and Welling, 2019), and permutations (Zaheer et al., 2017) and non-Euclidean domains, such as spheres (Cohen et al., 2018), point clouds (Fuchs et al., 2020) and graphs Satorras et al. (2021).

Our method is very similar to Bekkers (2019) proposing to parameterise filters on Lie groups using B-splines. A difference is that we are not limited to B-splines but generalise to arbitrary basis functions and consider filters in the context of relaxed equivariance.

**Relaxed equivariance**   Equivariance and invariance symmetry constraints have proven very useful in data modelling. However, hard symmetry constraints can be overly restrictive if data does not fully respect enforced symmetries (van der Ouderaa and van der Wilk, 2021). In digit-recognition, some robustness to rotational perturbation is desirable, whereas '6's and '9's may become indistinguishable under full rotational invariance. Likewise, Wang et al. (2022) improve real-world dynamics data modeling through relaxed symmetry constraints.

Some efforts have been made to parameterise approximate equivariance (Wang et al., 2022; Romero and Lohit, 2021). A recent generalisation of the convolution proposed by van der Ouderaa et al. (2022) allows for relaxed equivariance with explicit control over the amount of equivariance. We utilise this generalisation and demonstrate how our more efficient filter parameterisation can be effectively used to parameterise filters in this setting.

**Symmetry discovery**   Another interesting consequence of relaxing equivariance constraints, is that symmetry structure becomes differentiable and can thus be optimised with gradients. Recent work on automatic symmetry discovery (Benton et al., 2020; van der Ouderaa and van der Wilk, 2021; Immer et al., 2022) focus on invariances or data augmentation, which are easier to parameterise than layer-wise equivariance. We offer an efficient way to parameterise filters for relaxed equivariance, which may allow extensions of symmetry discovery methods to learn layer-wise equivariances.

## 3. Background

**Lie groups and Lie algebra**  A Lie group is a group with the structure of a differentiable manifold. To each Lie group $G$ we can associate a Lie algebra $\mathfrak{g}$: the tangent space of the Lie group at the identity element, capturing local structure of the Lie group. The Lie algebra always forms a vector space. The exponential map exp maps elements from the Lie algebra to the Lie group. As the Lie algebra is always a vector space, its elements $a \in \mathfrak{g}$ can be expanded in a basis $a = \sum_{i=1}^{\dim(G)} \alpha_i A_i$, with coefficients $\boldsymbol{\alpha} \in \mathbb{R}^{\dim(G)}$. We define the operator $a^\vee := \boldsymbol{\alpha} \in \mathbb{R}^D$ to encode Lie algebra elements as real vectors in the chosen basis. The exponential map $\exp : \mathfrak{g} \to G$ maps elements from the Lie algebra to the Lie group. We can also define a logarithm map $\log : G \to \mathfrak{g}$ that maps elements in the Lie group to the Lie algebra. Such a choice always exists, but it is not always smooth or unique.

**Equivariance and Invariance**  Equivariance is the property of a mapping such that transformations to the input result in equivalent transformations in the output. If changes are invertible, they can be described as the action of a group $G$ on some space $\mathcal{X}$. Formally, we say that a function $h : \mathcal{X} \to \mathcal{X}$ is *equivariant* to the group $G$ if $h(g \cdot x) = g \cdot h(x)$ for all $g \in G$, $x \in \mathcal{X}$. If the output of the function is independent to the action of the group G on the input, we say that the function is *invariant* to group $G$. Formally, a function $h : \mathcal{X} \to \mathcal{X}$ is invariant to group $G$ if $h(g \cdot x) = h(x)$ for all $g \in G$, $x \in \mathcal{X}$.

**Group convolutions**  Classical convolutions on real space encode translational equivariance. The group convolution $y$ encodes equivariance to other groups (Cohen and Welling, 2016):

$$y(v) = \int_G f(v^{-1}u)x(u)\mu_G(u) \qquad \hat{y}(v) = \frac{1}{N_{\mathrm{MC}}} \sum_{u \in \mathrm{nhbd}(v)}^{N_{\mathrm{MC}}} f(v^{-1}u)x(u) \qquad (1)$$

with convolutional kernel $f : G \to \mathbb{R}$, input $x : G \to \mathbb{R}$, and output $y : G \to \mathbb{R}$ all functions on group $G$ with Haar measure $\mu_G$. For compact groups, the equivariance constraint is both a sufficient and necessary condition for a linear map to be a convolution (Kondor and Trivedi, 2018). In other words, no equivariance without convolution (and vice versa). In practice, we can approximate the integral with $\hat{y}$ using $N_{\mathrm{MC}}$ Monte Carlo samples (Finzi et al., 2020).

**Relaxed equivariance**  If modeled data does not exactly follow a symmetry, strictly enforcing it is misspecified and approximate equivariance may be more desirable. We can quantify the amount of equivariance by the *equivariance error* (Wang et al., 2022):

$$||f||_{\mathrm{EE}} = \sup_{x,g} ||f(g^{-1}x) - g^{-1}f(x)|| \qquad (2)$$

for a function $f : G \to \mathbb{R}$. In case of strict equivariance, we have that $||f||_{\mathrm{EE}}=0$ and for relaxed equivariance we typically want $||f||_{\mathrm{EE}} < \epsilon$ for some $\epsilon$.

To relax equivariance, we use the construction proposed by van der Ouderaa et al. (2022) which uses filter functions $\widetilde{f_G} : G \times G \to \mathbb{R}$ in Eq. (8) that not only depends on the relative group element $v^{-1}u$, but also on an additional absolute input group element $u$. This additional argument breaks strict equivariance and the amount of equivariance can be controlled through this dependence.

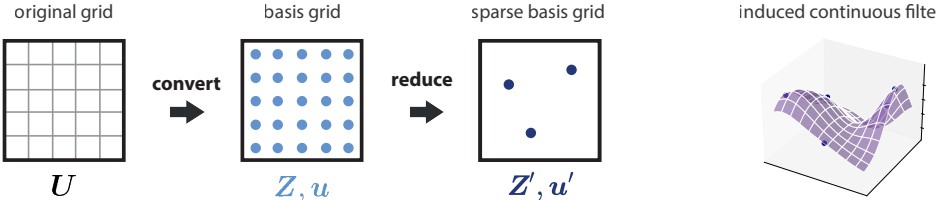

original grid     basis grid     sparse basis grid     induced continuous filter

**convert**    **reduce**

$U$      $Z, u$      $Z', u'$

Figure 1: Illustration of continuous basis function filter. Classical filters can be converted to continuous basis function filters and optionally further sparsified to reduce memory.

## 4. Continuous filters with basis functions

**Classical filters** Filters in classical discrete convolution $f : \mathbb{Z}^2 \to \mathbb{R}$ map discrete 2-dimensional filter space $\mathbb{Z}^2$ to the reals. Filter are typically only defined on a small local squared support $\mathcal{S} \subset \mathbb{Z}^2$ forming a grid which pixel coordinates can be summarised as row vectors in matrix $\boldsymbol{S} \in \mathbb{Z}^{S^2 \times 2}$. Filter weights can be stored in a square matrix $\boldsymbol{U} \in \mathbb{R}^{S \times S}$. As we treat filters as functions, we define it as a function $f(\begin{bmatrix} x_1 & x_2 \end{bmatrix}^T) = [\boldsymbol{U}]_{(x_1, x_2)}$ that indexes and returns weights within the support $[x_1, x_2]^T \in \mathcal{S}$, and 0 elsewhere. Instead of filters in discrete space $\mathbb{Z}^2$, we will now propose how to construct filters on continuous domains $\mathbb{R}^2$.

**Continuous filters with basis functions** Our goal is to parameterise smooth filter functions on $f : \mathbb{R}^2 \to \mathbb{R}$ that can be efficiently parameterised with a finite amount of memory. To do so, we consider a finite amount of $P$ anchor points, summarised in matrix $\boldsymbol{Z} \in \mathbb{R}^{P \times 2}$, to which we associate values listed in $\boldsymbol{u} \in \mathbb{R}^P$. In addition, we choose a basis function $\phi : \mathbb{R}^2 \times \mathbb{R}^2 \to \mathbb{R}$ and scalar noise parameter $\sigma^2$, and induce a continuous function $f$ from this finite set $(\boldsymbol{Z}, \boldsymbol{u})$ using simple basis function regression. This defines a continuous filter $f(\boldsymbol{x})$, which closed-form formula is given by Eq. (3), that can be evaluated at any arbitrary coordinate $\boldsymbol{x} \in \mathbb{R}^2$. The memory required to store the filter is no longer a function of filter size in $\mathcal{O}(S^2)$, but rather of the number of anchor points $\mathcal{O}(P)$.

**Converting from classical filters** The proposed basis function formulation differs from that of classical CNN filters as it enables continuous filter evaluations anywhere in $\mathbb{R}^2$. However, we can treat a classical filter as a special case of our basis function filter allowing for straightforward conversion between the two. To convert a classical discrete filters $f_{\text{classic}} : \mathbb{Z}^2 \to \mathbb{R}$ to a continuous basis function filters $f : \mathbb{R}^2 \to \mathbb{R}$ we can set our anchor point coordinates equal to the original sampling grid $\boldsymbol{Z} = \boldsymbol{S}$ and set the associated values equal to filter values $\boldsymbol{u} = \text{vec}(\boldsymbol{U})$, with zero noise $\sigma = 0$. This creates a basis function filter that has the exact same output at the original sampling grid, ie. $f(\boldsymbol{x}) = f_{\text{classic}}(\boldsymbol{x}) \quad \forall \boldsymbol{x} \in \mathbb{Z}^2$. In this case, the number of anchor points equals the number of pixels in sampling grid $P = S^2$, and the resulting memory complexity therefore remains equivalent. Transferring weights from regular convolutional layer to implementations of basis function filters can be of practical interest.

**Inference** The filter response at any location in $\boldsymbol{x} \in \mathbb{R}^2$ can be computed in closed-form:

$$f_{\boldsymbol{Z}, \boldsymbol{u}}(\boldsymbol{x}) = \boldsymbol{\phi}_{\boldsymbol{xZ}}^T (\boldsymbol{\Phi}_{\boldsymbol{ZZ}} + \sigma^2 \boldsymbol{I})^{-1} \boldsymbol{u}, \tag{3}$$

where $\boldsymbol{\phi}_{\boldsymbol{xZ}} := \begin{bmatrix} \phi(\boldsymbol{x}, \boldsymbol{z}_1) & \phi(\boldsymbol{x}, \boldsymbol{z}_2) & \cdots & \phi(\boldsymbol{x}, \boldsymbol{z}_P) \end{bmatrix}^T \in \mathbb{R}^P$ and $\boldsymbol{\Phi}_{\boldsymbol{zz}} \in \mathbb{R}^{P \times P}$ is defined by $[\boldsymbol{\Phi}_{\boldsymbol{ZZ}}]_{ij} := \phi(\boldsymbol{z}_i, \boldsymbol{z}_j)$ with $\boldsymbol{z}_i$ being the $i$'th row vector of $\boldsymbol{Z}$.

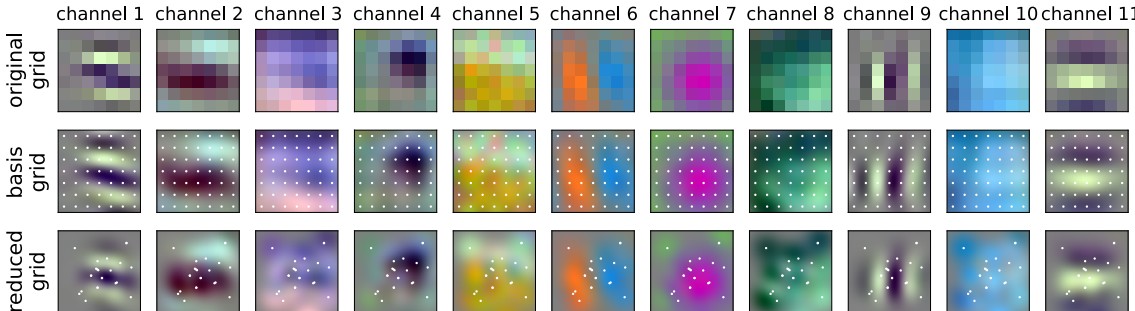

Figure 2: The first fifteen layer filters of a ResNet-34 pretrained on ImageNet (top) converted to continuous $\phi_{\mathrm{RBF}}$-basis filters that can be evaluated anywhere (middle). Basis function locations are not bound to a grid and can be sparsified (bottom).

**Train** The inference procedure $f_{\boldsymbol{Z},\boldsymbol{u}}(\boldsymbol{x})$ is differentiable w.r.t. basis function locations $\boldsymbol{Z}$, basis function values $\boldsymbol{u}$, and possible hyper-parameters in $\phi$. We can train basis function layers just like other neural network layers, treating $\boldsymbol{Z}$ and $\boldsymbol{u}$ as learnable parameters and optimise them together with model parameters using back-propagation.

**Sparsify** Optionally, we can further reduce the number of anchor points to $P' < P$ to a new set of locations $\boldsymbol{Z}' \in \mathbb{R}^{P' \times 2}$. We pick $\boldsymbol{Z}'$ with some scheme, such as random sampling or (row-wise) subset of $\boldsymbol{Z}$, and find new basis function values $\boldsymbol{u}' \in \mathbb{R}^{P'}$ that minimise the squared distance between new predictions $f_{\boldsymbol{Z}',\boldsymbol{u}'}(\boldsymbol{z}_i)$ and original basis functions:

$$\arg\min_{\boldsymbol{u}'} \sum_i ||f_{\boldsymbol{Z},\boldsymbol{u}}(\boldsymbol{z}_i) - f_{\boldsymbol{Z}',\boldsymbol{u}'}(\boldsymbol{z}_i)||^2. \tag{4}$$

We can find the following closed-form minimum norm solution for new anchor point values:

$$\boldsymbol{u}' = (\boldsymbol{A}^T\boldsymbol{A})^{-1}\boldsymbol{A}^T\boldsymbol{b}, \tag{5}$$
$$\text{with } \boldsymbol{A} = (\boldsymbol{\Phi}_{\boldsymbol{ZZ}'}(\boldsymbol{\Phi}_{\boldsymbol{Z}'\boldsymbol{Z}'} + \sigma^2\boldsymbol{I})^{-1}, \text{ and } \quad \boldsymbol{b} = (\boldsymbol{\phi}_{\boldsymbol{uZ}}^T(\boldsymbol{\Phi}_{\boldsymbol{ZZ}} + \sigma^2\boldsymbol{I})^{-1},$$
$$\text{where } \boldsymbol{\Phi}_{\boldsymbol{ZZ}'} := \phi(\boldsymbol{Z},\boldsymbol{Z}') \in \mathbb{R}^{P \times P'}, \text{ and } \quad \boldsymbol{\Phi}_{\boldsymbol{Z}'\boldsymbol{Z}'} := \phi(\boldsymbol{Z}',\boldsymbol{Z}') \in \mathbb{R}^{P' \times P'}.$$

Instead of only taking the arg min w.r.t. $\boldsymbol{u}'$, an even tighter fit may be obtained by optimizing both variables $\boldsymbol{Z}'$ and $\boldsymbol{u}'$. Unfortunately, there is no closed-form least-squares solution in this case, and would have to resort to other minimization strategies, such as gradient descent.

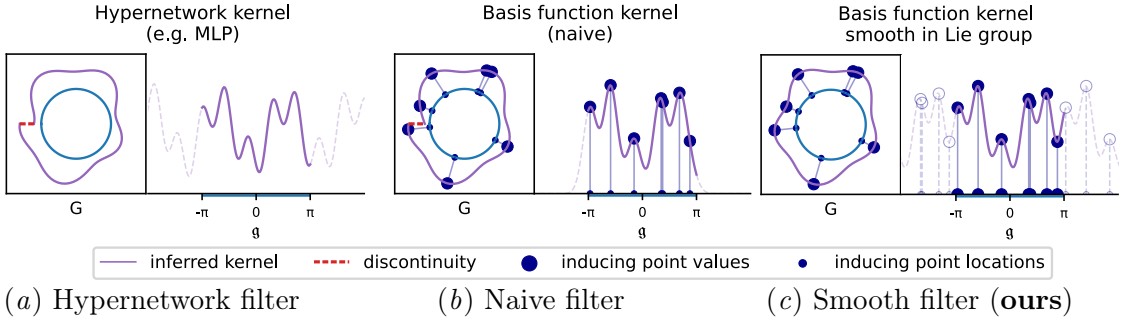

($a$) Hypernetwork filter      ($b$) Naive filter      ($c$) Smooth filter (**ours**)

Figure 3: Lie group filter parameterisations. (A) Smooth functions in Lie algebra are not necessarily smooth on the Lie group. (B) Naive basis function regression in the Lie algebra. (C) Proposed basis function filter efficient and smooth on SO(2).

## 5. Basis functions on Lie Groups

In the previous chapter, we discussed how to extend discrete convolutional filters on $\mathbb{Z}^2$ to continuous filters on $\mathbb{R}^2$. Now, we will further extend this to more general groups $f : G \to \mathbb{R}$, which can be useful to implement efficient group convolutions.

Again, we consider a finite amount of points but now place them on the group $Z = \{z_i\}_{i=1}^P \subset G$, and list the values associated to the points in $\boldsymbol{u} \in \mathbb{R}^P$. We can define a smooth function on the group $f : G \to \mathbb{R}$ from the finite set of anchor points and values $(Z, \boldsymbol{u})$ with basis function regression by defining a basis function that takes group elements as input $\phi_G : G \times G \to \mathbb{R}$. Now the difficulty lies in defining a basis function on the group. To do so, we choose to parameterise it in the Lie algebra $\mathfrak{g}$ associated to Lie group $G$:

$$\phi_G : G \times G \to \mathbb{R} : (g', g) \mapsto \phi(\log(g^{-1}g')^{\vee}) \tag{6}$$

where we choose some logarithm log map from group elements in $G$ to Lie algebra $\mathfrak{g}$, and use the $(\cdot)^{\vee} : \mathfrak{g} \to \mathbb{R}^{\dim(G)}$ operator to obtain a real vector that represents the group element in some chosen Lie algebra basis. Note that we take the difference between two elements in the Lie group $g^{-1}g'$ and that if two elements are the same $g=g'$ this maps to the identity element $\log(g^{-1}g)=\boldsymbol{0} \in \mathfrak{g}$ in the Lie algebra. This is desirable, as a more naive approach of mapping to the Lie algebra first, i.e. mapping to $\phi(\log(g') - \log(g))$, would depend on the location of the identity (see Fig. 3(b)). Using this construction we can define filters on the group in terms of regular basis functions in real vector space $\phi : \mathbb{R}^{\dim(G)} \to \mathbb{R}$, such as the $\phi_{\mathrm{RBF}}$.

This generalises Bekkers (2019) proposing the same technique for B-spline basis functions.

**Inference** Similar to basis function kernels for regular convolutions, we can find a value at any continuous location in filter space $g \in G$ in closed-form:

$$f_{\boldsymbol{Z},\boldsymbol{u}}(g) = \boldsymbol{\phi}_{gZ}^T(\boldsymbol{\Phi}_{ZZ} + \sigma^2\boldsymbol{I})^{-1}\boldsymbol{u}, \tag{7}$$

where we define $\boldsymbol{\phi}_{gZ} := \begin{bmatrix} \phi_G(g, z_1) & \phi_G(g, z_2) & \cdots & \phi_G(g, z_P) \end{bmatrix}^T \in \mathbb{R}^P$, define $\boldsymbol{\Phi}_{ZZ} \in \mathbb{R}^{P \times P}$ by $[\boldsymbol{\Phi}_{ZZ}]_{ij} = \phi_G(z_i, z_j)$, and a variance noise term $\sigma^2$ is added for numerical stability.

| Group | $N_{\mathrm{MC}}$ samples | Parameter matched | CIFAR-10 no augment | CIFAR-10 with augment | CIFAR-100 no augment | CIFAR-100 with augment |
|---|---|---|---|---|---|---|
| baseline (regular conv) | - | | 78.20 | 87.44 | 44.24 | 60.84 |
| p4 | 4 | | 77.57 | 89.83 | 44.18 | 65.28 |
| | | ✓ | 73.13 | 85.32 | 43.78 | 57.05 |
| SE(2) =T(2) ⋊ SO(2) | 4 | | 79.36 | 84.28 | 52.30 | 57.58 |
| | | ✓ | 76.47 | 78.06 | 48.29 | 57.18 |
| | 8 | | 83.06 | 90.17 | 55.40 | 64.65 |
| | | ✓ | 80.03 | 86.50 | 53.26 | 56.72 |
| | 16 | | 80.09 | 90.64 | 50.92 | 65.61 |
| | | ✓ | 81.57 | 88.43 | 53.87 | 60.93 |
| $\mathbb{R}^2 \rtimes \mathbb{R}^+$ | 3 | | 78.79 | 86.54 | 45.84 | 59.49 |
| | | ✓ | 78.03 | 84.70 | 48.61 | 54.69 |
| | 8 | | 78.78 | 86.89 | 46.02 | 60.18 |
| | | ✓ | 77.20 | 85.64 | 48.12 | 54.61 |

Table 1: Performance of group equivariant ResNet-18 with basis function filters. Test accuracy on CIFAR-10 and CIFAR-100 datasets for different Lie groups.

## 6. Basis functions for relaxed equivariance

In this work, we consider the method proposed by van der Ouderaa et al. (2022) to relax equivariance, letting convolutional filters depend on an additional absolute group elements:

$$\tilde{y}(v) = \int_G \widetilde{f_G}(v^{-1}u, u)x(u)\mu_G(u) \tag{8}$$

where filter $\widetilde{f_G} : G \times \widetilde{G} \to \mathbb{R}$, input $x : G \to \mathbb{R}$ and output $y : G \to \mathbb{R}$ are defined on group $G$. Unlike classic convolutions, filters $\widetilde{f_G}$ now operate on a group product space, taking not one but two group elements as input $v^{-1}u \in G$ and $u \in \widetilde{G}$. We can choose the relaxed subgroups through our choice of $\widetilde{G} \leq G$, and have $\widetilde{G}=G$ if all subgroups are relaxed. To emphasise the difference between regular convolutions and the operator of Eq. (8) that relaxes equivariance, we add a $\sim$-symbol to $\widetilde{y}$ and the filters $\widetilde{f_G} : G \times \widetilde{G} \to \mathbb{R}$ in case of relaxed equivariance. We proceed by defining continuous filters for relaxed equivariance by performing basis function regression on the group product space $G \times \widetilde{G}$. To do so, we consider a finite amount of points in this space $\widetilde{Z} = \{(z_i, \tilde{z}_i)\}_{i=1}^P \subseteq G \times \widetilde{G}$, with associated values listed as $\boldsymbol{u} \in \mathbb{R}^P$. Thus, this requires defining basis functions on the product space of the group product space:

$$\widetilde{\phi_G} : (G' \times \widetilde{G}') \times (G \times \widetilde{G}) \to \mathbb{R} : ((g', \tilde{g}'), (g, \tilde{g})) \mapsto \phi \left( \begin{bmatrix} \log(g^{-1}g')^\vee \\ \log(\tilde{g}^{-1}\tilde{g}')^{\tilde{\vee}} \end{bmatrix} \right) \tag{9}$$

where we choose a logarithm log-map from the Lie group $G$ to the Lie algebra $\mathfrak{g}$ and operators $(\cdot)^\vee : \mathbb{R}^{\dim(G)} \to \mathbb{R}$ and $(\cdot)^{\tilde{\vee}} : \mathbb{R}^{\dim(\widetilde{G})} \to \mathbb{R}$ to represent elements as real vectors by expanding in a chosen Lie algebra basis. We have defined basis functions on group product spaces in terms of used stationary basis functions in real vector space $\phi : \mathbb{R}^{\dim(G)+\dim(\widetilde{G})} \to \mathbb{R}$. Similar to Sections 4 and 5, we can construct filters for relaxed equivariance in closed-form:

$$\widetilde{f_G}(g, g') = \boldsymbol{\phi}_{(g,g')Z}^T (\boldsymbol{\Phi}_{ZZ} + \sigma^2 \boldsymbol{I})^{-1} \boldsymbol{u}, \tag{10}$$

where $\boldsymbol{\phi}_{(g,g')Z} := [\widetilde{\phi_G}((g', z_1'), (g, z_1)) \cdots \widetilde{\phi_G}((g', z_P'), (g, z_P)))]^T \in \mathbb{R}^P$, and define $[\boldsymbol{\Phi}_{ZZ}]_{ij} = \widetilde{\phi_G}((g', z_i'), (g, z_j))$ with shape $\boldsymbol{\Phi}_{ZZ} \in \mathbb{R}^{P \times P}$, and $\sigma^2$ for numerical stability.

**Note about smoothness in Lie group** One might ask whether filters on Lie groups defined in the Lie algebra (as in Sections 5 and 6) are still smooth on the Lie group. Even if the function in the Lie algebra is smooth, smoothness is not guaranteed on the Lie group. An example for SO(2) is depicted in Figs. 3(a) and 3(b). However, filters will still be smooth close to the identity and possible discontinuities are not likely to form problems if filters are only locally supported around the origin. This is typically true for classic convolutional filters, but is not often done for other groups, such as SO(2), in literature. This problem arises in many commonly used filter parameterisations, such as regular MLPs and encodings that use random Fourier feature spaces with non-integer frequencies, such as positional encodings or SIRENs Sitzmann et al. (2020). Our proposed basis function filters can still be guaranteed to be smooth on SO(2) by carefully choosing a periodic basis $\phi_{\text{PERIODIC}}$ (see Fig. 3(c)).

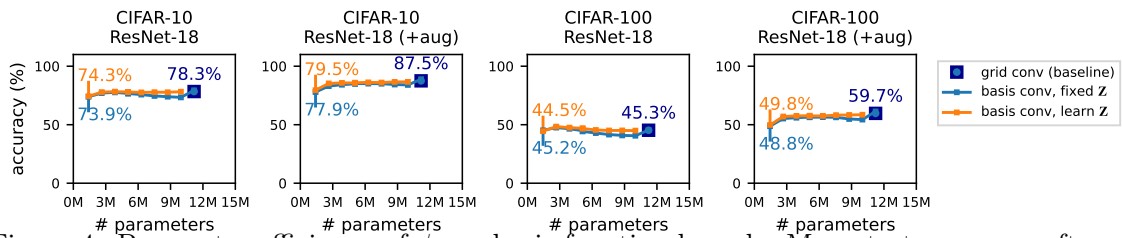

Figure 4: Parameter efficiency of $\phi_{\text{RBF}}$-basis function kernels. Mean test accuracy after reducing model parameters with procedure described in Eqs. (4) and (5) of Section 4.

## 7. Results

**Analysis of parameter efficiency**  To assess parameter efficiency of basis function filters we compare test accuracy for different number of anchor points $P$. The parameter count of basis function filters is directly proportional to $P$ and can be chosen freely, unlike the fixed parameter count of classical convolutional filters which correspond to the area of the chosen filter support (e.g. 25 parameters for a $5 \times 5$ filter). We compare test accuracy of regular convolutional layers ('grid convs') with basis function filters at varying parameter counts. We initialise equally in both cases and reduce the parameter counts before training using the sparsification procedure described in Eqs. (4) and (5) of Section 4, uniformly sampling anchor point locations within the filter support. Fig. 4 shows test accuracy against number of parameters on CIFAR data with and without data augmentation. We demonstrate a reduction of model parameters up to a factor of $\sim 10\times$ with a few percentage points drop in test accuracy in the most extreme case: 3-4% without data augmentation and 8-9% with data augmentation. Surprisingly, we have merely a single $P{=}1$ anchor point left in this setting. Per layer, that amounts to a single anchor point location shared over channels and a single anchor point value parameter for each filter channel. More details in App. C.

**Large equivariant networks with basis function kernels**  The parameter-efficiency of basis function filters allow us to construct group equivariant versions of large commonly used convolutional architectures at parameter counts equivalent to regular non-equivariant models. To assess effectiveness, we compare test accuracy of such equivariant versions for different groups: regular discrete translation, discrete rotations $p4$, continuous planar roto-translations SE(2), and continuous planar scalings $\mathbb{R}^2 \rtimes \mathbb{R}^+$. Table 1 shows test accuracy for models trained with and without data augmentation. Equivariant networks for rotation groups outperform the regular baseline for SE(2) given enough (typically $>8$) Monte Carlo samples $N_{\mathrm{MC}}$. We do not observe such improvement for planar scaling and hypothesise that could be due to sampling error of small filters. This would be an issue with sampled regular representations in general and not of our basis function parameterisation as such. Overall, we demonstrate that basis function filters can effectively be used to construct equivariant versions of large commonly used convolutional architectures. Embedding group symmetries into the architecture yields higher test accuracies, as expected.

**Relaxed equivariance and symmetry discovery**  Lastly, we investigate the use of basis function filters to parameterise relaxations of equivariance. We consider roto-translation $G{=}$SE(2) and consider different amounts of relaxation of subgroup SO(2) by varying $\widetilde{\boldsymbol{\omega}}$. Basis function filters allow us to repeat the cross-validation experiments in van der Ouderaa et al. (2022) on large commonly used ResNet architectures. We directly report test accuracy in the upper part of Table 2. Instead fixing the amount of relaxation, we also consider learning $\widetilde{\boldsymbol{\omega}}$ from training data by treating it as a parameter and optimising it with backpropagation together with the model parameters. As $\widetilde{\boldsymbol{\omega}}$ directly controls layer-wise equivariance constraints, this allows gradient-based learning of symmetry constraints. As suggested by van der Ouderaa et al. (2022), we add a regularizing $\lambda||\widetilde{\boldsymbol{\omega}}||^2$ to the cross-entropy loss that encourages $\widetilde{\boldsymbol{\omega}}$ to be low and thus encourages stricter symmetry constraints. This is analogous to the regularisation proposed in Augerino (Benton et al., 2020) to learn invariances through data augmentations and similarly requires tuning of $\lambda$. We report test accuracy for different values for $\lambda$ in the bottom part of Table 2. We observe that learned equivariance constraints $\widetilde{\boldsymbol{\omega}}$

achieve similar performance to the best fixed setting for $\widetilde{\omega}$ and outperform baselines in most cases. This suggests that our efficient sparse filters can be an efficient parameterisation for relaxed equivariance. Like most symmetry discovery methods, we still need to parameterise the class of symmetries that can be learned and limit ourselves to learning the amount of rotational equivariance to the SO(2) subgroup in an SE(2)-equivariant network. Also, the used regularisation requires tuning an additional hyperparameter. Immer et al. (2022) observe that selecting the right amount of invariance with regularisation used in Augerino can be difficult and proposes an alternative loss motivated by marginal likelihood approximations. Investigating such objectives could improve layer-wise equivariance learning in combination with the proposed filter parameterisation is interesting future work. Regardless, our experiments indicate that basis function filters can effectively parameterise filters for relaxed equivariance in large networks. We improve over van der Ouderaa et al. (2022) which had to consider smaller architectures due to added hypernetwork parameters and we recommend using the proposed filter parameterisation instead.

| $SE(2) = T(2) \rtimes \widetilde{SO(2)}$ | | | CIFAR-10 | | CIFAR-100 | |
|---|---|---|---|---|---|---|
| | # params | fixed $\widetilde{\omega}$ | no augment | with augment | no augment | with augment |
| baseline | 11.2 M | - | **80.03** | **86.50** | **53.26** | **56.72** |
| relaxed equivariance | 11.2 M | 0 | 79.46 | 85.97 | 53.49 | 57.47 |
| | 11.2 M | 0.0001 | 79.46 | 85.97 | 53.49 | 57.47 |
| | 11.2 M | 0.01 | 83.09 | 88.53 | 57.64 | **61.29** |
| | 11.2 M | 0.1 | **84.16** | **89.20** | **57.89** | 61.09 |
| | 11.2 M | 0.5 | 80.26 | 86.95 | 52.02 | 58.30 |
| | 11.2 M | 1.0 | 75.35 | 82.07 | 44.69 | 52.15 |
| | 11.2 M | 1.5 | 71.89 | 78.01 | 43.35 | 47.08 |
| | 11.2 M | 2.0 | 66.09 | 64.94 | 39.81 | 38.32 |
| | | learned $\widetilde{\omega}$ | | | | |
| | 11.2 M | $\lambda = 0.00001$ | 83.34 | 89.12 | 57.92 | 60.28 |
| | 11.2 M | $\lambda = 0.00005$ | 83.41 | 88.84 | **58.72** | **60.32** |
| | 11.2 M | $\lambda = 0.0001$ | **83.51** | **89.17** | 58.18 | 60.15 |

Table 2: Learning the amount of equivariance per layer. We use layers with relaxed equivariance for fixed and learned amounts of equivariance constraints controlled by $\omega'$.

## 8. Conclusion

This work proposes a practical parameterisation for continuous (group) convolutional filters with sampled regular representations. Existing methods often rely on MLP hypernetworks which may require many additional model parameters compared to classical filters. Instead, we propose to infer continuous filters from a small finite set of anchor points utilising basis function regression. The approach is general and enables parameter-efficient continuous filters in regular convolutional networks (CNNs), group-equivariant convolutional layers (G-CNNs) and filters for relaxed equivariance. We assess parameter efficiency of basis function filters at reduced parameter counts. Futhermore, the proposed basis function filters allow us to construct equivariant versions of large commonly used convolutional neural network architectures that can outperform regular baselines on CIFAR-10 and CIFAR-100 classification tasks. Furthermore, the approach allows for relaxations of equivariance without increasing the parameter count. In this last setting, we demonstrate efficient symmetry discovery by automatically learning layer-wise equivariance constraints from training data.

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

## Appendix A. Additional related work

**Basis function regression**    Basis function regression is a well known procedure[1] which we use to infer values $\boldsymbol{y}^* \in \mathbb{R}$ at arbitrary continuous points in $D$-dimensional space $\boldsymbol{x}^* \in \mathbb{R}^D$ from a finite number of $P$ points in $\boldsymbol{X} \in \mathbb{R}^{P \times D}$ with known values $\boldsymbol{y} \in \mathbb{R}^P$. We use a basis function $\phi(\boldsymbol{x}', \boldsymbol{x}) : \mathbb{R}^D \times \mathbb{R}^D \to \mathbb{R}$ and perform inference in closed-form:

$$\boldsymbol{y}^* = \boldsymbol{\phi}_{\boldsymbol{x}^* \boldsymbol{X}}^T (\boldsymbol{\Phi}_{\boldsymbol{X}\boldsymbol{X}} + \sigma^2 \boldsymbol{I})^{-1} \boldsymbol{y} \tag{11}$$

where vector $\boldsymbol{\phi}_{\boldsymbol{x}^* \boldsymbol{Z}} \in \mathbb{R}^P$ and matrix $\boldsymbol{\Phi}_{\boldsymbol{Z}\boldsymbol{Z}} \in \mathbb{R}^{P \times P}$ are constructed using the basis function $[\boldsymbol{\phi}_{\boldsymbol{x}\boldsymbol{Z}}]_i := \phi(\boldsymbol{x}, \boldsymbol{Z}_i)$ and $[\boldsymbol{\Phi}_{\boldsymbol{Z}\boldsymbol{Z}}]_{ij} := \phi(\boldsymbol{Z}_{i:}, \boldsymbol{Z}_{j:})$ and regularizing observational variance scalar $\sigma^2$. Here $\boldsymbol{Z}_{i:} \in \mathbb{R}^D$ denotes the i'th row of $\boldsymbol{Z}$. If $\phi(\boldsymbol{x}', \boldsymbol{x})$ can be rewritten as $\phi'(\boldsymbol{x} - \boldsymbol{x}')$, the basis function is called 'stationary'. In this case, we will simply write the basis function as the one-argument function $\phi : \mathbb{R}^D \to \mathbb{R}$. One of the most commonly used basis functions $\phi$ in regression is the stationary squared exponential radial basis function $\phi_{\text{RBF}}(\boldsymbol{a}) = \sigma^2 \exp\left(-2\omega^2 \boldsymbol{a}^T \boldsymbol{a}\right)$, with scalar variance $\sigma^2 \in \mathbb{R}$ and frequency $\omega^2 \in \mathbb{R}$.

## Appendix A. Implementation

**Basis function factorisation**    We choose to factor the basis functions as $\phi(\boldsymbol{a}) = \prod_{i=1}^D \phi_i(a_i)$ with $\boldsymbol{a} \in \mathbb{R}^D$ and represent individual dimensions using radial or periodic basis functions:

$$\phi_{\text{RBF}}(a_i) = \exp\left(-\frac{1}{2}\omega_i^2 a_i^2\right) \qquad \phi_{\text{PERIODIC}}(a_i) = \exp\left(-2\omega_i^2 \sin^2\left(\frac{\pi}{2}|a_i|\right)\right) \tag{12}$$

where we set $\phi_i = \phi_{\text{PERIODIC}}$ for dimensions that correspond to the SO(2) subgroup and $\phi_i = \phi_{\text{RBF}}$ for dimensions that correspond to translation $\mathbb{R}^2$ or scale $\mathbb{R}^+$ subgroups. Note that we have $D = \dim(G)$ for strict equivariance and $D = \dim(G) + \dim(\widetilde{G})$ for relaxed equivariance. Factorising basis functions across dimensions differs from factorising the filter itself across dimensions, which is often called a 'separable filter' (Knigge et al., 2022). Separable filters have efficiency benefits, but limit the functions a filter can represent. Unlike separable kernels, our proposed factorisation of basis functions still yield filters that can approximate any function, given enough anchor points.

**Intuition behind frequency parameters**    To understand the effect of frequency parameters on filters, it can be helpful to separately consider the first frequencies, overloading notation $\boldsymbol{\omega} := [\omega_1, \ldots, \omega_{\dim_G}]^T \in \mathbb{R}^{\dim(G)}$, and the remaining $\dim(\widetilde{G})$ frequency components as $\widetilde{\boldsymbol{\omega}} \in \mathbb{R}^{\dim(\widetilde{G})}$.

The frequencies $\boldsymbol{\omega}$ control the spectral properties of regular filters. For example, frequencies for translation $G = \text{T}(2)$ would be two-dimensional $\boldsymbol{\omega} \in \mathbb{R}^2$ and influence the respective frequencies along the x- and y-axis of the two-dimensional filter. On the other hand, the additional frequencies for relaxed equivariance $\widetilde{\boldsymbol{\omega}}$ control the amount of equivariance of relaxed subgroups, where higher frequencies correspond to more relaxation. Conversely, setting $\widetilde{\boldsymbol{\omega}} = \boldsymbol{0}$ corresponds to no relaxation in which case the layer becomes equivalent to a regular (group) convolution (van der Ouderaa et al., 2022), which is guaranteed to be strictly equivariant $||f||_{\text{EE}} = 0$.

---

1. Basis function regression is more commonly known as 'kernel ridge regression', where $\phi$ is the kernel function. To avoid confusion with convolutional kernels, we do not use this name.

**Group elements as real vectors** We map Lie group elements to Lie algebra elements through a choice of log and expand them through $(\cdot)^\vee$ or $(\cdot)^{\widetilde{\vee}}$ in a Lie algebra basis so they can be implemented as real vectors. The choice of basis is somewhat arbitrary and can be set according to the application. We choose our basis such that a length of 1 corresponds to the apothem of a sampled filter support for T(2), doubling in scale for $\mathbb{R}^+$ and a full rotational orbit for SO(2). For relaxed equivariance, the same basis is used except for T(2) where group elements correspond to the input domain and therefore the apothem of the input image support is used instead of that of the sampled filter support. We always initialise frequency components as 1, if not specified otherwise.

**Example for relaxed rotational equivariance** In our experiments, we will consider relaxing the SO(2) subgroup of SE(2) = T(2) $\rtimes$ SO(2). In this case, we have that $\dim(G) = \dim(\mathrm{T}(2)) + \dim(\mathrm{SO}(2)) = 2 + 1 = 3$ and $\dim(\widetilde{G}) = \dim(\mathrm{SO}(2)) = 1$. Therefore, $D = \dim(G) + \dim(\widetilde{G}) = 3 + 1 = 4$. We will use $\phi_1 = \phi_2 = \phi_{\mathrm{RBF}}$ for the dimensions corresponding to translation T(2) subgroup, and $\phi_3 = \phi_4 = \phi_{\mathrm{PERIODIC}}$ for the dimensions that correspond to rotation SO(2) and relaxation of rotation SO(2) subgroup. Thus, the frequency vectors in this case are of dimensionality $\boldsymbol{\omega} \in \mathbb{R}^{\dim(G)} = \mathbb{R}^3$ and $\widetilde{\boldsymbol{\omega}} \in \mathbb{R}^{\dim(\widetilde{G})} = \mathbb{R}^1$.

## Appendix B. Appendix: Training details

### B.1. Architecture

For the architectures, we used the standard ResNet-18 and ResNet-152 architecture He et al. (2016) implementations of PyTorch Paszke et al. (2017) with the following parameter counts.

| Model | # Parameters |
|---|---|
| ResNet-18 (10 classes) | 11,181,642 (11.2 M) |
| ResNet-18 (100 classes) | 11,227,812 (11.2 M) |

Table 3: Parameter count of used ResNet architecture

### B.2. Dataset details

We evaluate with the CIFAR-10 and CIFAR-100 datasets (Krizhevsky et al., 2009). The CIFAR-10 dataset consists of 60,000 32x32 colour images. CIFAR-100 consists of 60,000 32x32 colour images in 100 classes. For both datasets, we use the default 50,000/10,000 split for training and testing and apply standard zero mean and unit standard deviation normalisation using training set statistics.

### B.3. Optimization details

In all experiments, we optimise using Adam (Kingma and Ba, 2014) with an initial learning rate of 0.001 cosine decayed to zero over 400 epochs.

## Appendix C. Parameter reduction with sparsified convolutions

Unlike grids of regular convolutions, the $P$ basis coordinates (rows of $\mathbf{Z}$) of basis function kernels are no longer required to lie in a grid. Furthermore, the number of basis functions per filter can be chosen arbitrarily and can be lower than the number of pixel grids in regular convolutions. In Eqs. (4) and (5) of Section 4, we formulate a procedure to sparsify parameters of convolutional kernels, similar to 'pruning'. Note that we reduce the number of parameters that describe the convolutional filters and (and thus reduce storage cost), but the size of the sampled filter during training/inference remains the same. In Section 7, we evaluate parameter efficiency of basis function kernels by evaluating test performance of a ResNet-18 model with reduced number of basis function coordinates. In Tables 4 and 5, we report more detailed results of this experiment including exact parameter counts of reduced models as well as test accuracy. Mean and standard error are reported over 3 random seeds.

| | # params per filter | | | ResNet-18 w/ $\mathbf{Z}$ fixed | | | ResNet-18 w/ $\mathbf{Z}$ free | | |
|---|---|---|---|---|---|---|---|---|---|
| reduce factor | $P'_{1\times1}$ | $P'_{7\times7}$ | $P'_{3\times3}$ | # params total | test acc | test acc (+aug) | # params total | test acc | test acc (+aug) |
| baseline 1.0 | 1 | 9 | 49 | 11,689,552 | **79.38** ±0.19 | **88.89** ±0.12 | - | - | - |
| reduce 0.9 | 1 | 44 | 8 | 10,467,984 | 73.35 ±0.80 | 83.99 ±0.43 | 10,468,334 | 78.21 ±0.69 | 86.64 ±0.18 |
| 0.8 | 1 | 39 | 7 | 9,246,416 | 73.94 ±0.45 | 84.35 ±0.29 | 9,246,724 | 77.72 ±0.40 | 86.58 ±0.16 |
| 0.7 | 1 | 34 | 6 | 8,024,848 | 74.59 ±0.39 | 85.04 ±0.30 | 8,025,114 | 77.79 ±0.57 | 86.03 ±0.17 |
| 0.6 | 1 | 29 | 5 | 6,803,280 | 75.74 ±0.81 | 85.13 ±0.46 | 6,803,504 | 77.60 ±0.53 | 86.18 ±0.44 |
| 0.5 | 1 | 24 | 4 | 5,581,712 | 76.64 ±0.90 | 85.12 ±0.24 | 5,581,894 | 78.00 ±0.67 | 85.90 ±0.37 |
| 0.4 | 1 | 20 | 4 | 5,580,944 | 76.70 ±0.69 | 84.97 ±0.22 | 5,581,118 | 77.96 ±0.52 | 85.71 ±0.27 |
| 0.3 | 1 | 15 | 3 | 4,359,376 | 77.65 ±0.52 | 84.17 ±0.28 | 4,359,508 | 78.27 ±0.19 | 85.73 ±0.23 |
| 0.2 | 1 | 10 | 2 | 3,137,808 | 77.04 ±0.24 | 83.01 ±0.18 | 3,137,898 | 77.86 ±0.38 | 85.28 ±0.26 |
| 0.1 | 1 | 5 | 1 | 1,916,240 | 73.87 ±0.35 | 77.92 ±0.56 | 1,916,288 | 74.31 ±0.29 | 79.50 ±0.41 |

Table 4: Test accuracy of sparsified networks on CIFAR-10. Test accuracy of ResNet-18 model trained with reduced number of parameters, with and without data augmentation. Total parameter count and parameter count per filter are reported, as well as standard error over three seeds $\frac{\sigma}{\sqrt{(3)}}$.

| | # params per filter | | | ResNet-18 w/ $\mathbf{Z}$ fixed | | | ResNet-18 w/ $\mathbf{Z}$ free | | |
|---|---|---|---|---|---|---|---|---|---|
| reduce factor | $P'_{1\times1}$ | $P'_{7\times7}$ | $P'_{3\times3}$ | # params total | test acc | test acc (+aug) | # params total | test acc | test acc (+aug) |
| baseline 1.0 | 1 | 9 | 49 | 11,689,552 | **45.44** ±0.06 | **61.39** ±0.05 | - | - | - |
| reduce 0.9 | 1 | 44 | 8 | 10,467,984 | 40.57 ±0.56 | 44.95 ±0.64 | 10,468,334 | 54.37 ±0.16 | 58.52 ±0.10 |
| 0.8 | 1 | 39 | 7 | 9,246,416 | 40.86 ±0.70 | 45.10 ±0.58 | 9,246,724 | 54.89 ±0.31 | 58.34 ±0.32 |
| 0.7 | 1 | 34 | 6 | 8,024,848 | 41.59 ±0.71 | 45.07 ±0.27 | 8,025,114 | 56.31 ±0.61 | 58.16 ±0.17 |
| 0.6 | 1 | 29 | 5 | 6,803,280 | 42.86 ±0.66 | 45.51 ±0.87 | 6,803,504 | 56.48 ±0.81 | 57.54 ±0.86 |
| 0.5 | 1 | 24 | 4 | 5,581,712 | 44.27 ±0.93 | 46.73 ±0.69 | 5,581,894 | 56.34 ±0.30 | 57.48 ±0.31 |
| 0.4 | 1 | 20 | 4 | 5,580,944 | 44.85 ±0.81 | 46.90 ±0.34 | 5,581,118 | 56.49 ±0.18 | 57.69 ±0.50 |
| 0.3 | 1 | 15 | 3 | 4,359,376 | 46.54 ±0.43 | 47.50 ±0.51 | 4,359,508 | 55.98 ±0.44 | 57.56 ±0.33 |
| 0.2 | 1 | 10 | 2 | 3,137,808 | 47.53 ±0.57 | 48.38 ±0.16 | 3,137,898 | 55.28 ±0.16 | 56.99 ±0.37 |
| 0.1 | 1 | 5 | 1 | 1,916,240 | 45.18 ±0.35 | 44.47 ±0.09 | 1,916,288 | 48.77 ±0.49 | 49.80 ±0.32 |

Table 5: Test accuracy of sparsified networks on CIFAR-100. Test accuracy of ResNet-18 model trained with reduced number of parameters, with and without data augmentation. Total parameter count and parameter count per filter are reported, as well as standard error over three seeds $\frac{\sigma}{\sqrt{(3)}}$.

