# OpenReview forum: "Sparse Convolutions on Lie Groups"
_NeurIPS.cc/2022/Workshop/NeurReps — NeurReps 2022 Poster_

### Official Review · Reviewer_jMDh · 2022-10-12
**A promising path to pare problematic parameter prices of (partially-) equivariant networks**

**Confidence:** 5
**Soundness:** 4
**Presentation:** 4
**Contribution:** 4
**Overall Rating:** 8

**Summary:**

In 'Sparse Convolutions on Lie Groups', the authors exhibit a novel technique to reduce the parameter complexity of neural networks that use basis functions to implement group convolutions. The approach uses basis functions of the form $\phi(g_1, g_2)$, where $g_i$ are associated with elements of the group. The approach is to pick a subset of $P$ points in $G$ and compute the filter response at a new location $x$ as $\sum_{i=1}^{P}\phi(g_i, x)\gamma(g_i)$, where the coefficients $\gamma(g_i, x)$ can be chosen specifically so that the evaluations of the filter function at some other points is approximately the same as the original basis function that required more parameters. They authors apply their technique to classification problems CIFAR-10 and CIFAR-100 on images and present tradeoffs between the complexity of their models (both in parameters and # monte carlo samples) for the groups p4, SE(2), and 2D translations + scale. Lastly they applied their technique to an approximately-equivariant architecture of Ouderaa et al

**Questions:**

1) was the gradient descent technique for jointly picking the P points and their values (as opposed to the exact min-norm technique of e.g. page 4) empirically tested or was this just a suggestion?

2) Is the idea of the technique described by the equation on page 4 essentially interpolation? Is the $\sigma^2 I$ a random sample from a gaussian distribution? Or if not, why is $\simga^2$ notation used?

3) in figure 3 -- what is naive basis function regression referring to? Is this the technique where $P$ = total number of original points (ie no compression)?

4) table 1: please clarify what does parameter matched mean?

5) "Overall, we demonstrate that basis function filters can effectively be used to construct equivariant versions of large commonly used convolutional architecture. Embedding group symmetries into the architecture yields higher test accuracies, as expected" -> This is very cool. Are you able to re-use the convolutional kernels learned for the non-equivariant model when performing this procedure (at least, for certain groups)? Also, is this claim justified by table 2? If so please clarify that this is when using the approximate and not exact equivariance. If it is not justified by table 2 please clarify where is the evidence for this claim that "embedding group symmetries into the architecture yields higher test accuracies, as expected"

6) (also mentioned above) what is pseudo point? what is pseudo about them? I suggest replacing this with a different term (see suggestions above)

**Limitations:**

Sticking with symmetry invariance is a reasonable choice but there are multiple natural equivariant tasks (e.g. orientation prediction) which are equivariant and not just invariant. Only exploring the invariant tasks is a limitation of this work.

A central claim by the authors seems to be that prior methods for using basis functions (either with MLP or with the 'naive regression') would result in models with parameter complexity that is too high to be practically relevant. This claim seems plausible to me; however, it is not backed up with a specific benchmark (e.g. in the time to evaluate one forward pass of a network using ABC technique vs XYZ technique)

It would be easier to read if the captions on the figures were more self-contained and didn't require reading all the surrounding sections of text. This would improve the accessibility of the experiments which are a highlight of the work.

Maybe it's explained in the abstract, which I just skimmed -- but is the source code going to be released? It would be great to share the source code and ideally also the pretrained equivariant resnet weights please

**Recommended Decision:**

3: Accept

**Relevance:**

4: Highly relevant

**Strengths And Weaknesses:**

I think the coolest idea in this paper seemed to me that you could take an existing convolutional network architecture (e.g. ResNet) and "equivarify" it (I made up that term and it is not used by the authors) without increasing the model complexity too much. As strict equivariance often comes with a high complexity to train and evaluate, this is an important contribution and may help make of equivariant models in more practically relevant. The idea of basis function compression is original and well-motivated and I think should be of interest to the community. it is impressive to make an equivariant version (even approximately so) of a large well-used traditional convolutional architecture such as a resnet.

It would have been cool if the authors had performed truly equivariant/covariant regression in addition to group-invariant classification. The technique is only used for real-valued functions as applied to group convolutions and it's unclear whether other functions on a group can be compressed in this way, however this is not directly relevant to the motivation from equivariance. It would be neat if the authors explored more the choice of basis functions and how this impacts performance. However, the experiments are a strength of this paper overall and it is reasonable to leave such questions to (hopefully) be answered by future work.

I put an 'accept' vote however this is conditional on small revisions to address a number of smaller comments about the manuscript which I will include below. I hope they help in polishing the manuscript so that it is camera-ready.

1) The term "psuedo-point" is not one I am familiar with, it is never defined and I would suggest to replace it with something more descriptive like "reference elements" or "anchor points in the Lie algebra". If the authors insist on using the terminology "pseudo-point", they should please define what it means at the first instance in which they use it and in particular please explain what is "pseudo" about these points.

2) in the abstract: "architectural structure neural networks" -> "architectural structure of neural networks"

3) page 3 "just a few pseudo-point" -> "just a few pseudo-points"
3) Bottom of page 1 -- "parameter efficiency and overall performance" please clarify what is meant by "overall performance" -- is it related to the throughput in a computational / FLOPs sense or is performance = classification accuracy?

4) page 2: "exact in the infinite width limit" -> "exactly in the infinite width limit"

5) page 2: "points clouds" -> "point clouds"

6) Although the method is agnostic of the particular choice of basis function, the authors should mention what basis functions they use in their experiments, though I would find it acceptable to refer to the appendix for the details of the exact choice of functions. I was also curious, since some of their fitting techniques require a matrix inverse computation, is there some completeness or rank property needed for this to be well-defined? I.e., can any function whatsoever be used or are they reliant upon particular properties of the (e.g. RBF) basis functions.

7) on page 2, nitpick: use the backtick (`) to make a left apostrophe for the '6's and '9's

8) page 3: "lie algebra always is a " -> "lie algebra is always a"
9) page 3: "enforcing strictly enforcing" -> "strictly enforcing"
10) equation (2) evaluates to zero and is therefore clearly incorrect, perhaps there is supposed to be a $g^{-1}$ on the outside of the function evaluation

11) in the equation on page 4 should explain why the matrix is always (or, w.h.p. ) invertible

12) please edit the plots in figure 2 so that the axes titles are legible (larger text) -- note you could replace "channel 1 channel 2" etc. with "channel:    1  2 3 4" and then use larger text. Also it would be fine to show fewer of the filters to convey an idea of how well the approximation works

13) please clarify in figure 2 the details of the choice of basis function

14) in figure 3: "naively basis function regression" -> "naive regression of basis function" (I think?). Please also add more details to explain what is conveyed by figure 3.

15) page 6: 'basis functions kernels' -> 'basis function kernels'

16) table 1 would be easier to read if it were more like table 2 where the best-in-class models are boldfaced

17) in equation 9, what is G'? (is it just a copy of G?) please explain this in the text

18) please add reference(s) for SIRENs or else define what they are
19) figure 4: it is not clear what this is conveying and it is difficult to make out the behavior due to its gentle slope. I suggest adjusting the lower and upper range of these plots to be 45% and 90%, respectively, so that the variation is magnified. Also please clarify which basis functions are used for this experiment.
20) page 9: "effectivity" -> "effectiveness"

21) " large commonly used convolutional architecture." ->  "  large commonly used convolutional architectures."


**Submission Track:**

Proceedings Paper (9 Page)

---

### Official Review · Reviewer_6QAV · 2022-10-16
**Interesting approach but some clarifications needed**

**Confidence:** 3
**Soundness:** 3
**Presentation:** 2
**Contribution:** 3
**Overall Rating:** 5

**Summary:**

The paper presents a method for expressing the kernels in a convolution model more compactly in terms of a sparse set of support points.  The main idea is nicely illustrated in Figure 1.  However the method or formula by which one arrives a the smooth continuous filter shown at right from the support points is unclear.  Is there some functional form you are fitting?  if so, what is it?
Overall the paper would benefit from more a more clear and straightforward explanation and derivation of the method.


**Questions:**

How is equation (3) arrived at?
What is the meaning of phi(x,z_i)?  what do the two arguments refer to?
Figure 2 is nice, but I have a feeling this hides some complexity.  You express the filter as a few support points as opposed to many discrete samples, but isn't this hiding the computational cost of computing the smooth function from the support points?  Also each of the support points needs to be encoded with a coordinate, whereas in the pixel representation it is implicit by its index within the weight vector.  So when you consider all these factors together, is there really a savings in the end?


**Limitations:**

see question above about actual computational cost of the method.


**Recommended Decision:**

3: Accept

**Relevance:**

3: Solid fit

**Strengths And Weaknesses:**

Strength:  a potentially useful way to reparameterize convnets
Weakness:  paper is somewhat unclear, see questions below


**Submission Track:**

Proceedings Paper (9 Page)

---

### Official Review · Reviewer_tcfR · 2022-10-19
**Basis Function approach or LIe Group CNNs**

**Confidence:** 4
**Soundness:** 3
**Presentation:** 3
**Contribution:** 3
**Overall Rating:** 7

**Summary:**

The paper considers the problem of designing neural networks that incorporate (full or partial) symmetries induced by Lie groups. The theory for such equivariant networks is well-established, although building efficient versions remains an active area of research. One method of implementing such networks parameterizes continuous filters by means of small shallow NNs/hypernetwork MLPs, which take the coordinates as input and output the filter response. The fact that such parameterizations aren't efficient is the main motivating point for this paper, which in turn considers a basis function approach. That is, the kernels are parameterized by pseudo-points instead (this seems related to, but also different from the basis-based parameterizations provided in Weiler & Cesa, for instance), which are regressed over. The work closest in spirit to the proposed approach is the B Splines approach of Bekkers (2019), and also has relevance for partial equivariance and also symmetry discovery.

The main contribution of the paper is explicated in section 4. It is discussed how classical filters (for 2D) can be extended from Z^2 to R^2, meaning for any point in R^2 we would like a continuous filter ouput. More specifically, the authors consider a finite-set of pseudo-points, summarize them in a matrix, and lift them to a continuous parameterization by using a basis function approach (equation 3), which is actually standard in functional data analysis. This construction is then extended to the case of Lie groups, using pseudo points in a similar manner. The basis functions are parameteized using the associated Lie algebra. This formulation also allows for smooth integration with some recent work on relaxed equivariance (e.g. van der Ouderaa et al), which is covered in section 6.

A set of experiments analyze:
- Parameter efficiency of the basis function filters (for different number of pseudo points). It is shown that model parameters can be reduced significantly without much loss in accuracy.
- Accuracy of these models (and different versions thereof) is reported in table 1. Table 2 shows additional results with relaxed equivariance.

**Questions:**

Minor comments:
- Please make inline citations in the intro and throughout the paper. A lot of statements are made as factual, but without any accompanying references. Some of these statements are far from obvious.
- Typo in line 2, section 1: should be "resulting in.."
- Typo in line 4, section 1: "Extensions to other groups exists" --> "Extensions to other groups exist"
- Typoe, section 1: "with just a few pseudo-point" --> "with just a few pseudo-points
[lots of other typos, please correct]

**Limitations:**

Somewhat

**Recommended Decision:**

3: Accept

**Relevance:**

4: Highly relevant

**Strengths And Weaknesses:**

- The papers builds on some recent work on relaxed equivariance and efficient parameterizations of Lie group CNNs, and uses a basis function approach to parameterize filters.
- The approach permits incorporating relaxed equivariance, and also symmetry discovery.
- Results are somewhat (understandbly) preliminary, but indicate the usefulness of the method proposed both in terms of parameter efficiency and accuracy.

**Submission Track:**

Proceedings Paper (9 Page)

---

### Decision · Program_Chairs · 2022-10-21

Accept (Poster)